# Trans-Tokenization and Cross-lingual Vocabulary Transfers: Language Adaptation of LLMs for Low-Resource NLP

**François Remy***
Internet and Data Science Lab (IDLab)
Ghent University & imec (BE)
francois.remy@ugent.be

**Pieter Delobelle***
Department of Computer Science
KU Leuven & Leuven.AI (BE)
pieter.delobelle@kuleuven.be

**Hayastan Avetisyan**
German Centre for Higher Education
Research and Science Studies (DE)
avetisyan@dzhw.eu

**Alfiya Khabibullina**
AI & Data Bootcamp Ghent 2024
BeCode asbl (BE)
alfiyamkhabibullina@gmail.com

**Miryam de Lhoneux**
Department of Computer Science
KU Leuven (BE)
miryam.delhoneux@kuleuven.be

**Thomas Demeester**
Internet and Data Science Lab (IDLab)
Ghent University & imec (BE)
thomas.demeester@ugent.be

## Abstract

The development of monolingual language models for low and mid-resource languages continues to be hindered by the difficulty in sourcing high-quality training data. In this study, we present a novel cross-lingual vocabulary transfer strategy, trans-tokenization, designed to tackle this challenge and enable more efficient language adaptation. Our approach focuses on adapting a high-resource monolingual LLM to an unseen target language by initializing the token embeddings of the target language using a weighted average of semantically similar token embeddings from the source language. For this, we leverage a translation resource covering both the source and target languages. We validate our method with the Tweeties, a series of trans-tokenized LLMs, and demonstrate their competitive performance on various downstream tasks across a small but diverse set of languages. Additionally, we introduce Hydra LLMs, models with multiple swappable language modeling heads and embedding tables, which further extend the capabilities of our trans-tokenization strategy. By designing a Hydra LLM based on the multilingual model TowerInstruct, we developed a state-of-the-art machine translation model for Tatar, in a zero-shot manner, completely bypassing the need for high-quality parallel data. This breakthrough is particularly significant for low-resource languages like Tatar, where high-quality parallel data is hard to come by. By lowering the data and time requirements for training high-quality models, our trans-tokenization strategy allows for the development of LLMs for a wider range of languages, especially those with limited resources. We hope that our work will inspire further research and collaboration in the field of cross-lingual vocabulary transfer and contribute to the empowerment of languages on a global scale.

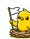 We release our models at https://huggingface.co/Tweeties &
a Python library at https://github.com/LAGoM-NLP/transtokenizer.

---

*Shared first-authorship

# 1 Introduction

Multilingual tokenization is unfair, with all existing approaches inadvertently favoring some languages over others (Petrov et al., 2023; Rust et al., 2021). This bias is particularly pronounced in multilingual subword tokenization techniques, which face the impossible task of distributing their token capacity equitably among all supported languages. Western European languages often benefit from this, thanks to their shared alphabet and linguistic heritage (Limisiewicz et al., 2023). Although character or byte-level encoders appear to handle diverse scripts more fairly, they frequently struggle to capture meaningful word-level information, especially in non-ideographic languages with limited alphabets (Libovický et al., 2022; Edman et al., 2022). Furthermore, byte-level tokenizers also display bias due to the substantial disparities in unicode encoding efficiency across languages.

In light of these challenges, we stress the need for a more personalized approach, where each language is equipped with its own tokenizer, specifically tailored to its unique needs. Unfortunately, the challenge of developing monolingual language models for all the world's languages has never been more present due to the vast amounts of data required to train large language models (LLMs), as evidenced by the technical reports of Mistral (2023), OLMo (2024) and Gemma (2024). The trillion tokens required for training LLMs simply does not exist in most languages (Joshi et al., 2020), turning transfer learning into a requirement.

Moreover, serving a wide array of monolingual LLMs at scale remains impractical. Efficient computation necessitates the batch-processing of requests (Pope et al., 2022), but many languages also suffer from intermittent workloads. This also makes it unsustainable to dedicate extensive GPU resources to continuously host often-idling LLMs, while the time required to load them back into memory impedes many commercial applications that require low latency (Alizadeh et al., 2024).

In this paper, we introduce several key innovations designed to democratize the training and deployment of high-quality monolingual models across a diverse set of languages. More specifically, we demonstrate how model conversion enables researchers to adapt LLMs to new languages using a very limited amount of resources, with a performance competitive with continual pre-training. Our approach preserves most layers of the original model, thereby facilitating the batch-processing of queries written in different languages, a critical factor in making the deployment of language-specific models economically viable.

# 2 Background and related work

The adaptation of pre-trained language models (PLMs) to new languages and domains remains a key challenge in the field of NLP. A promising approach to address this challenge is vocabulary transfer, as the technique involves replacing the vocabulary of a PLM with one that is more aligned with the target language or domain.

Vocabulary transfer has been explored as a means to adapt models to new linguistic contexts without the need for extensive retraining. Gee et al. (2022) demonstrated the efficacy of this approach in compressing language models, showing that it not only improves the performance of domain-adapted models by increasing their effective context size but also reduces their memory footprint and inference time by eliminating unused tokens.

Further investigating the impact of vocabulary transfer, Mosin et al. (2023) focused on the role of corpus-specific tokenization in the fine-tuning of transformer models. They suggest that combining corpus-specific tokenization with vocabulary transfer can accelerate the adaptation process and enhance model performance, thanks to a better tokenization.

Despite these findings, the process of tokenizer swapping often necessitates the reinitialization and retraining of the embedding table, resulting in substantially degraded performance. To address this issue, researchers have explored methods to preserve as much of the original model's embeddings as possible, especially when the source and target tokenizers share morphosemantic similarities (Artetxe et al., 2020; Garcia et al., 2021; Gogoulou et al., 2022). However, the application of these methods is limited by the availability of shared tokens and the morphosemantic proximity between the involved languages (de Vries et al., 2021).

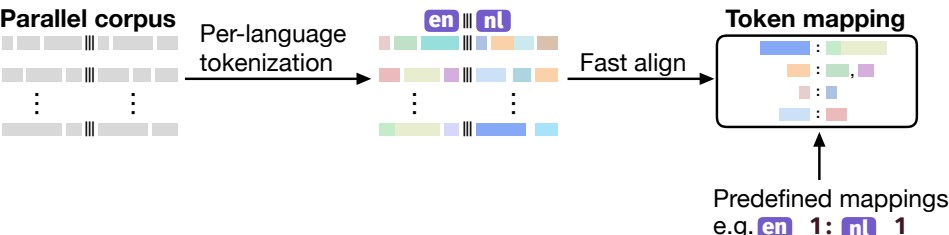

(a) **Token alignment** is performed first based on a tokenized parallel corpus using a SMT-based alignment tool, to establish a probabilistic token mapping. We provide snippets of each stage of the full pipeline in Appendix E.

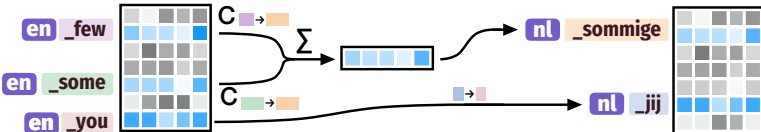

(b) **Embedding mapping** is then performed, as the embedding table for the target language (e.g. Dutch, indicated by nl) is initialized from the embeddings of mapped tokens in the source language (e.g. English, indicated by en), while preserving hidden layers.

Figure 1: **Overview of our Trans-Tokenization method**

An alternative approach involves the use of embedding alignment techniques to generate embeddings for a new language based on those trained for another model (Kalinowski & An, 2020). While promising, this strategy faces a significant challenge: many languages lack the high-quality LLMs necessary for the alignment, creating a "chicken-and-egg" problem.

In response to these challenges, several authors (Minixhofer et al., 2022; Remy et al., 2023), concurrently proposed a novel strategy for efficient language adaptation through cross-lingual embedding initialization. By leveraging bilingual character n-gram embeddings, this approach facilitates the cross-lingual mapping of tokens, showing particular promise for models with large tokenizers (BERT-style models) and language pairs with semantically-related character n-grams. However, our follow-up experiments indicated that this approach performed less well for GPT-style models and unrelated language pairs (see Appendix D).

Building on the above-mentioned works, we introduce a new cross-lingual vocabulary transfer strategy, named trans-tokenization. This approach is designed to facilitate the adaptation of GPT-style LLMs to languages with distinct scripts and linguistic families, addressing the limitations of existing methods and expanding the potential for language model adaptation across a broader spectrum of languages.

## 3 Trans-Tokenization

Tokenizers limit the range of languages that a model can effectively support. Even when performance for an unseen language is acceptable, tokens that are used to encode words in this language often map to smaller subwords, reducing the effective context length, and are rarely trained properly. As a consequence, the embeddings for these tokens are also less meaningful, even between languages with a shared ancestral language or with significant language contact.

For instance, the English word 'music' was borrowed from the French 'musique', which is encoded by two tokens ('mus' + 'ique') in most English BPE-based tokenizers. However, the 'mus' token is unlikely to have been pretrained well, since the token 'music' exists.

To address this problem, we intuitively want to create a mapping between tokens based on a translation scheme, instead of relying on orthographic or morphological similarity. However, subword tokenization prevents the direct use of word translation dictionaries.

To achieve this mapping, our Trans-Tokenization method therefore relies on two steps, as shown in Figure 1: (i) a token alignment generated using a parallel corpus and (ii) an embedding mapping. Depending on the model, there is also a third step, where the untied language modeling head undergoes the same mapping as the embedding table.

**Token alignment:** We start by tokenizing both sides of a parallel corpus using either the source or target tokenizer, but re-encode words as single units[1] for non-ideographic languages. Next, we pass this tokenized parallel corpus through a Statistical Machine Translation (SMT) model, FastAlign by Dyer et al. (2013). FastAlign provides a probabilistic token mapping based on the real-world evidence extracted from the parallel corpus (e.g. revealing that the Dutch token ˍvijftien is matched with ˍfifteen about 52% of the times, with ˍ15 about 46% of the times, and with ˍFif ... teen the remaining 2% of the times).

Because SMT-based alignment sometimes results in incorrect alignments, we discard any token alignment whose count is smaller than 10 (this can be increased for larger corpora). This ensures that the final mapping stays readable, avoiding a long tail of noisy mappings.

To deal with tokens whose mapping does not require real-world evidence (e.g. numbers, special characters, . . . ), we predefine a set of additional one-to-one mappings, which are implemented as a plus-one smoothing, just in case this mapping never appears in the parallel corpus. We also perform this operation to align internal tokens such as CLS.

Since we rely on a word-level SMT alignment, adjustments need to be made for words which are split into multiple tokens by either tokenizer (e.g. matching ˍFif + teen with ˍVijf + tien). Two strategies are used to address this. One consists in considering that every token from the target word is matched with every token of the source word (*all-to-all mapping*). This strategy makes no assumption, but is a bit wasteful. The other strategy relies on the token order within words, matching the first token of the target word with the first token[2] of the source word (*in-order-mapping*). This strategy assumes that order is preserved across languages, which is not always true. However, a generative model needs to know which token is the first in a word; when all tokens are initialized with the same average, the model cannot determine which token should come first. In practice, we average the results of both calculations to obtain an adjusted per-token count ($C_{s \to t}$) from the word alignment.

**Embedding mapping:** The second step of our method is to initialize the embeddings in the target model with their respective embeddings from the source model. For some tokens, this is relatively straightforward, as there is only one translation (e.g. 'ˍyou' in Figure 1b). However, this is not always the case. When a token has multiple possible translations, the embedding of these translations are averaged proportionally to the number of times the alignment appeared in the parallel corpus ($C_{s \to t}$), as illustrated in Appendix E.

**Language modeling head mapping:** When trans-tokenizing LLMs for which the language modeling head is not tied with the input embeddings, we apply the same mapping on the language modeling head as well (which projects the hidden dim to the vocab size).

We analyze deeper one such mapping, between Armenian and English, in Appendix K.

---

[1] We determine word boundaries using the definition of 'letter' in unicode (\p{L}) and tokenization: mergeable tokens lack an initial whitespace (ˍtoken) or start with a word continuation sign (##izer), depending on the tokenizer. We perform SMT alignment at a word-level instead of at a token level, since tokens often occur in multiple words. Preliminary experiments showed that the mappings obtained from using tokens without re-merging were of lower quality, with more noise. If needed, we split up word mappings back to individual tokens in the next stage, where we map the embeddings.

[2] When the lengths do not match, tokens are matched proportionally to their relative position (e.g. for 2-vs-3, the first target token would be matched partially to the first and second source tokens, with token match counts of respectively ⅔ and ⅓ of the initial word match count, thus preserving the total).

## 4    Hydra language models

After adapting an English language model to a new language using the method described above, we can also leverage our mapped embedding space to create models which accept tokens from both tokenizers. We refer to these models as 'Hydra' LLMs, in reference to their ability to stand on multiple legs (embedding tables) and grow multiple (language modelling) heads. These Hydra LLMs can be utilized for tasks such as the translation of texts or instructions from the source language to the target language, by encoding the source language using the initial tokenizer and producing new tokens in the target language using the newly-trained tokenizer. This approach is analogous to code-switching.

We envision several configurations of Hydra LLMs in this article, but focus our experiments to the zero-shot cross-lingual translation from the source language to the target language, as we believe this task to be the most promising and the most reliably measurable using well-established metrics. To test our hypothesis, we extend the popular Transformers library from HuggingFace (Wolf et al., 2020) by introducing a new `LlamaHydraForCausalLM` class.

The most important difference of Hydra models lies in the usage of distinct input and output vocabularies; while the input vocabulary includes the output vocabulary, it also contains one or several other embedding tables used to support tokens from other languages. To use the embeddings located beyond the main tokenizer, an offset can be added to the token ids produced by the additional tokenizers. To perform back-propagation, the labels of tokens located beyond the output vocabulary should be set to a masked value (e.g. -100).

We hypothesize (but did not verify) that the two bottom layers of the source model should probably be used to encode tokens from the original vocabulary instead of the layers finetuned for the target language. However, in our experiments, only the weights of the trans-tokenized model are used for inference, as this did not seem to cause any issue.

## 5    Experimental setup

In the next sections, we discuss the performance of our method for several languages, with a focus on low-resource (§ 6.1, § 6.2, § 6.3, § 6.4) and mid-resource languages (§ 6.5, § 6.6).

To test the capabilities of our transfer learning method in a worst-case scenario, we decided to evaluate it on Tatar, an endangered low-resource language which has few similarities with English. Indeed, 75% of the 8070 languages encoded in URIEL (Littell et al., 2017) are more similar to English than Tatar. This figure remains identical if we only consider the 184 languages featuring a two-letter code, as a proxy for language prominence.

Additionally, none of the 10 languages supported by our translation model at initialization feature an URIEL similarity of more than 35% with Tatar (as a comparison point, English has a 40% similarity with Korean and 28% with Chinese). Finally, there is only a limited amount of training data for the language. For example, Tatar Wikipedia contains only 1.43% as many articles as English Wikipedia (68th out of 278 languages).

We also evaluate trans-tokenization performance on Armenian, an Indo-European language with distinctive characteristics, such as an entirely unique writing script and the absence of any closely related languages within its sub-group. Being closer to English, we expect to see better results in Armenian than in Tatar for language modeling tasks (lower perplexity).

Finally, we wrap up our evaluations with Dutch, a Western Germanic language very close to English, and for which more resources are available, enabling to test more conclusively the capabilities of our models in factuality and reasoning. We also finetune our Dutch model using a Chat dataset, to compare its capabilities with other existing models.

Our evaluations cover a wide range of tasks, ranging from classical language modeling to language understanding and text summarization techniques for low-resources languages, and extending to more advanced SQuAD-type evaluations for our mid-resources languages. We also evaluate our Hydra LLMs using zero-shot translation from English to Tatar, a challenging language pair for which no high-quality dataset exists.

## 5.1 Trans-Tokenization Experiments

For our low-resource and mid-resource experiments, we train several models and baselines. We start from Mistral-7B (Team MistralAI et al., 2023), as this is a high-quality model for English. We also perform some ablation studies in the low-resource setting to understand which initialization and finetuning approach works best. To keep the result tables compact, the strategies used for training these models are detailed below:

**Mistral** We use the target language in the prompt with Mistral (2023), without finetuning. This strategy relies on the original model's pre-existing understanding of the language from its training corpus. While effective for well-resourced languages, it is unlikely to yield good results for low-resource languages due to limited data exposure during pre-training. Nevertheless, for languages with more resources like Dutch, the source model provides a solid baseline.

**Mistral+FT** We perform continual pre-training using the original tokenizer of the language model. Although BPE tokenizers are universal encoders (Sennrich et al., 2016), most merged tokens cater to prominent languages, resulting in inefficient encoding for low-resource ones.

**MistralRAND** We reinitialize the embedding table and language modeling head, retraining them using the in-domain corpus. While effective for high-resource languages, this strategy leads to substantially degraded performance for low-resource languages.

**MistralAVG** As an improvement over the preceding strategy, we restore the embedding of tokens shared between the source and target tokenizers. For Tatar, this concerns only around 12% of the tokens. The embeddings of all remaining tokens are then initialized with the average of the previously-mapped embeddings (to keep them in distribution).

**WECHSEL** We apply WECHSE (Minixhofer et al., 2022) to the languages in our setup, we initialize the embeddings of tokens using a bilingual dictionary derived from our SMT-aligned corpus. For Dutch, we test WECHSEL with (i) the original bidirectional dictionary and (ii) an equal-sized dictionary derived from our SMT-aligned corpus. For Tatar, we only follow the latter strategy. We do this since the original dictionaries were of extremely low quality: the Dutch one contained approximately 50% inaccurate or completely wrong translations[3] and the Tatar dictionary contains mostly text in the wrong language and script, making any comparison unfair.

**Tweety** Finally, we apply our trans-tokenization to initialize the embedding tables, as introduced in Section 3, to improve transfer learning by providing initialization for most tokens based on a cross-lingual token alignment. This strategy yields good results across the board and we refer to the resulting models as *Tweety* (Appendix M).

For low-resource languages, we use the language-specific split of OSCAR-2301 as training data (Ortiz Suárez et al., 2019; Abadji et al., 2022), and train a new 32k BPE tokenizer on it. To keep cross-lingual batch-processing possible, we finetune only the embedding table, the language modeling head, and the top two and bottom two layers of the Transformer, keeping the remaining 28 layers frozen. This choice of layers can be justified by their close proximity to the embedding layers, and the analyses of multilingual LLMs by Wendler et al. (2024) and Zhao et al. (2024). All models are trained using the same compute: 41M tokens with all layers frozen, and 66M tokens with the top 2 and bottom 2 layers unfrozen. These experiments run in fewer than 10 hours on a A100 GPU.

For our mid-resource language experiments, we report results for the full finetuning of the models over 400M tokens sourced from the C4 corpus (Raffel et al., 2019). This took less than a day on 2 A100 GPUs. We compare our model with Mistral-7B (Team MistralAI et al., 2023), the model we started from, GPT NEO 1.3b Dutch (Havinga, 2024) as well as related works (Minixhofer et al., 2022; Dobler & de Melo, 2023; Lin et al., 2024). A complete description of all experiments can be found in Appendix L.

---

[3]We release our bidirectional dictionaries on `https://github.com/LAGoM-NLP/transtokenizer/blob/master/notebooks/export/alignments/`.

## 5.2 Hydra Experiments

For low-resource translation experiments, we use the following Hydra LLMs:

**- HydraTower**: We apply trans-tokenization to the TowerInstruct model (Alves et al., 2024), initializing Tatar tokens by averaging mappings from English-Tatar and Russian-Tatar parallel corpora. For this, we use the No-Language-Left-Behind corpora (NLLB et al., 2022). We report in Appendix J our analysis on the benefits of multi-language initialization.

**- HydraTower+BackFT**: The previous model is further finetuned for the translation task using back-translation (Poncelas et al., 2018), with 2.2Mb of Tatar passages as expected output and 1.4Mb of English pseudo-translation provided by Google Translate as input.

We compare our LLMs with the only two publicly available English-to-Tatar MT systems:

**- Google Translate**: Tatar support was added in 2020 along with 4 other languages.

**- Microsoft Translator**: Tatar support was added in 2021 along with 11 other languages.

We also compare with the base **TowerInstruct** model (both before and after finetuning on the same parallel data used for initializing the trans-tokenization).

# 6 Evaluations

## 6.1 Low-Resource Language Modeling

The first way in which we evaluate the model adaptation strategies is by reporting the validation perplexity of the trained models. To ensure a fair comparison between models having different tokenizers, we report the "per native token" perplexity (that is, we normalize the perplexity reported by our library relative to the number of tokens required to represent a Tatar text using the tokenizer of the model; as detailed by Mielke (2019)).

| Model | Perplexity | | Train Tokens |
|---|---|---|---|
| **Mistral** | 60.38 | exp(3.1321) * 8116/3081 | 0M |
| **Mistral+FT** | | | |
| *(2x2 layers + embed.)* | 11.43 | exp(1.4681) * 8116/3081 | 107M |
| *(embeddings only)* | 14.25 | exp(1.6881) * 8116/3081 | 41M |
| **MistralRAND** | | | |
| *(2x2 layers + embed.)* | 80.74 | exp(4.3913) | 107M |
| *(embeddings only)* | 205.35 | exp(5.3247) | 41M |
| **MistralAVG** | | | |
| *(2x2 layers + embed.)* | 17.05 | exp(2.8361) | 107M |
| *(embeddings only)* | 25.11 | exp(3.2232) | 41M |
| **WECHSEL** improved dict. | | | |
| *(2x2 layers + embed.)* | 11.67 | exp(2.4569) | 107M |
| *(embeddings only)* | 31.40 | exp(3.4467) | 41M |
| **Tweety-7b-tatar-v24a** (ours) | | | |
| *(2x2 layers + embed.)* | **10.96** | exp(2.3947) | 107M |
| *(embeddings only)* | 19.69 | exp(2.9802) | 41M |

Table 1: **Perplexity for the Tatar language**, compared with the normalized perplexities of our baselines and ablation studies. Similar results for Armenian can be found in Appendix K along with a qualitative analysis of its SMT mapping.

## 6.2 Low-Resource Language Understanding

We also evaluate our adaptation strategies with the 1-shot performance of generative models on the SART Word Analogies dataset (Khusainova et al., 2023), and comparing them with the existing word embedding baselines. Despite looking trivial, this task remains quite challenging in a 1-shot setting due to the lack of instruction.

| Model | Accuracy | Model | Accuracy |
|---|---|---|---|
| Mistral | 23.25 | SkipGram | 23.45 |
| Mistral+FT | 25.42 | FastText | 18.11 |
| MistralRAND | 0.00 | GloVe | 17.48 |
| MistralAVG | 17.00 | Google Translate: | |
| **Tweety-7b-tatar-v24a** (ours) | **49.34** | Mistral+GTrans | ∼44.10 |

Table 2: Accuracy of models on Tatar the semantic word analogies from the SART dataset. Refer to Appendix F for a detailed scoring per analogy type, and analysis thereof.

In the case of Mistral+GoogleTranslate, translation inaccuracies affect the final results; to reduce the impact, the accuracy for this model pair was computed based on the English translations of the Tatar input. In practice, the answer would likely have to be translated back into Tatar, further reducing the quality of the answer. This was only done to get a rough idea of how well an English model would score on this task, in English.

## 6.3 Low-Resource Text Summarization

The 1-shot text summarization task is the third way we use to evaluate our Tatar models. We compute ChrF (Popović, 2015) to compare the generated summaries and the reference. We report our results in Table 3 and a description of the eval corpus in Appendix G.

| Model | ChrF against reference | Standard deviation |
|---|---|---|
| Mistral | 13.30 | (std = 0.27) |
| Mistral+FT | 23.15 | (std = 0.20) |
| MistralRAND | 3.79 | (std = 0.36) |
| **Tweety-7b-tatar-v24a** (ours) | **30.03** | (std = 0.28) |
| Mistral+GTrans | **30.43** | (std = 0.20) |

Table 3: Textual similarity of generated summaries with a reference. The Mistral + Google Translate results score the similarity of the Tatar translation of the Mistral summary of an English translation of the Tatar input.

## 6.4 Low-Resource Machine Translation

To evaluate our Hydra models, we focus on three English-to-Tatar machine translation tasks: two experiments relying on our text summarization dataset, as well as one smaller-scale evaluation on short social media messages scraped from Mastodon. For the latter, we paid a professional translator to provide high-quality references. Refer to Appendix H for a more detailed description of the datasets.

For the long text translation task, we showcase the advantage of using LLMs in translation systems, providing the gold standard of the short text as a 1-shot example in the prompt, to perform neural fuzzy repair (Bulte & Tezcan, 2019, +NFR in Table 5).

The translations of the 125 social media messages were also ranked pairwise by one of the authors, a native Tatar speaker. When no translation was good enough, neither received a preference vote. The professional translation won 51 pairwise votes, Google Translate 29, HydraTowerFT 24, and Microsoft Translator 10.

This confirms that HydraLLMs are competent machine translation systems.

| Model | Short Text | | Long Text | | Social Media | |
|---|---|---|---|---|---|---|
| RandomInDistrib | 17.8 | ±0.1 | 15.3 | ±0.6 | 16.7 | ±0.9 |
| TowerInstruct | 17.5 | ±0.4 | 13.5 | ±0.3 | 17.2 | ±0.5 |
| TowerInstruct+ParFT | 24.5 | ±0.4 | 16.5 | ±0.3 | 20.6 | ±0.6 |
| HydraTower+ParFT | 39.6 | ±0.5 | 18.4 | ±0.5 | 33.1 | ±1.4 |
| HydraTower | 47.3 | ±0.4 | 32.8 | ±0.4 | 39.2 | ±1.5 |
| HydraTower+BackFT | 53.7 | ±0.2 | 33.6 | ±0.3 | 46.1 | ±1.4 |
| Microsoft Translator | 54.9 | ±0.2 | 33.8 | ±0.4 | 48.7 | ±1.0 |
| Google Translate | **55.5** | ±0.2 | 35.3 | ±0.2 | ~~63.8~~ | ±1.8 |
| HydraTower+BackFT+NFR | —— | —— | **39.2** | ±0.6 | —— | —— |

Table 4: **Machine translation scores (ChrF) between texts and their reference translations**. Social medial references were produced by a professional translator in Tatarstan. The Google Translate results on this set are striked-through because of a possible data contamination, see Appendix I. RandomInDistrib refers to the average score obtained by comparing random pairs of texts from the reference sets, and serves as an absolute baseline. ParFT refers to finetuning the model on the parallel data used to initialize the Hydra embeddings. BackFT refers to finetuning the model on a small but high-quality set of Tatar text back-translated to English using Google Translate.

## 6.5 Mid-Resource Language Modeling

For evaluating our method on a mid-resource language, we train a Dutch model for 40 GPU hours and 417M tokens (see Appendix L for all details), we first compute a validation perplexity on the 'tiny' subset of the Dutch section of C4 (Raffel et al., 2019).

We trans-tokenize Mistral-7B (Team MistralAI et al., 2023) to use the vocabulary of GPT NEO 1.3b Dutch (Havinga, 2024) to make an easy comparison between both models, especially since we also train on the same dataset. Despite a significantly lower number of training tokens (417M versus 33B), our model obtains a perplexity of 11.1, compared to GPT NEO with 21.2. Mistral-7B has a lower perplexity, but there are fewer tokens and the tokenizer is not adapted to Dutch, meaning that more words are needed and the per-token perplexity is lower (Mielke, 2019). Based on the evaluation tokens counts, 33.1% more tokens are needed.

We also compare to related works in the mid-resource setting, more specifically WECH-SEL (Minixhofer et al., 2022), FOCUS (Dobler & de Melo, 2023) and MaLA-500, an adaptation of Llama 2 for 534 languages (Lin et al., 2024). For WECHSEL, we test the two variations of the (i) original bidirectional dictionary and (ii) an improved one based on our token mapping, as explained in § 5.1.

| | Tokenizer | | Training | Normalized |
|---|---|---|---|---|
| Model | Type | $|\mathcal{V}|$ | tokens | PPL |
| `mistral-7b-v0.1` | English BPE | 32,000 | 6-8T | *9.4* |
| WECHSEL (Minixhofer et al., 2022) | Dutch BPE | 50,257 | +0.4B | 34.3 |
| + improved Dutch dictionary | | | +0.4B | 27.1 |
| FOCUS (Dobler & de Melo, 2023) | Dutch BPE | 50,257 | +0.4B | 31.9 |
| `tweety-7b-dutch-v24a` (ours) | Dutch BPE | 50,257 | +0.4B | 11.1 |
| `gpt-neo-1.3b-dutch` | Dutch BPE | 50,257 | 33B | 21.2 |
| `mala-500-10b-v2` | Multilingual BPE | 260,164 | +30-60B | *18.9* |
| `tweety-7b-dutch-v24a` (ours) | Dutch BPE | 50,257 | +8.5B | **7.7** |

Table 5: **Test-set perplexity of Dutch models.** To make a fair comparison, we normalize italicized perplexities to our tokenizer, as described by Mielke (2019). We group models with the same tokenizer and evaluate our model at two checkpoints (0.4B and 8.5B tokens).

## 6.6 Mid-Resource Language Understanding

In addition to this intrinsic evaluation, we also ran a language understanding benchmark, SQuAD-NL (Rajpurkar et al., 2018), which is one of the evaluations of ScandEval (Nielsen, 2023) that was translated to Dutch. We compare our model to Mistral-7B and GPT NEO 1.3b Dutch. Additionally, we evaluate TowerBase-7B, a multilingual model supporting Dutch and which has been pre-trained for 20B more tokens starting from Llama 2 (Touvron et al., 2023). We observe that our model performs best in the one-shot and two-shot settings, but not the 0-shot setting where its answers are not always compatible with the SQuAD format.

| Model | Tokenizer | | SQuAD-NL ACC | | |
| | Type | $|\mathcal{V}|$ | 0-shot | 1-shot | 2-shot |
|---|---|---|---|---|---|
| mistral-7b-v0.1 | English BPE | 32 000 | **14.3** | 21.3 | 24.2 |
| towerbase-7b-v0.1 | English BPE | 32 000 | 13.0 | 20.9 | 22.6 |
| gpt-neo-1.3b-dutch | Dutch BPE | 50 257 | 0.0 | 0.0 | 0.0 |
| tweety-7b-dutch-v24a (ours) | Dutch BPE | 50 257 | 9.0 | **25.8** | **27.6** |

Table 6: **Dutch Language Understanding Evaluations.**

## 7  Discussion

**Advantages over other approaches.**   As our results demonstrate, our language adaptation method is capable of producing high-quality LLMs for low-resource languages, at a fraction of the cost of training similar-sized LLMs from scratch, and improved performance over continual pretraining. Unlike massively multilingual models, which inherently create tokenization unfairness, our trans-tokenized monilingual models offer each language an equal share of the embedding budget. This could in turn enable layer-sharing between languages. Finally, our work demonstrates that evidence-based SMT mappings perform better than traditional character-based embedding reinitialization techniques.

**HydraLLMs.**   Hydra LLMs extend the trans-tokenization concept to enable zero-shot cross-lingual tasks. In English-to-Tatar translation (Table 4), the HydraTower model performs competitively with commercial systems, and can further benefit from high-quality finetuning. This demonstrates the potential of Hydra LLMs for low-resource machine translation without extensive high-quality parallel data, leveraging the strengths of large language models in cross-lingual scenarios. However, our setup only enables translation in the High-to-Low resource direction, as our finetuning causes the LLM to lose its fluency in the source language. We leave the investigation of the reverse direction as future work.

## 8  Conclusion

In this work, we have introduced a novel approach to the adaptation of LLMs for low-resource languages through cross-lingual vocabulary transfers. Our experiments with the Tweeties series of trans-tokenized LLMs and Hydra LLMs have demonstrated the effectiveness of our approach across a range of downstream tasks and languages.

Notably, the development of a state-of-the-art machine translation model for Tatar, achieved in a zero-shot manner with Hydra LLMs, underscores the potential of our strategy to make significant strides in language technology for languages that have historically been underrepresented in NLP research.

We hope that our contributions will inspire further exploration and innovation in the field, and that the limitations we mentioned in Appendix A will be addressed in future works, some of which we already suggest in Appendix B. We are eager to read your works!

**Author Contributions**

All authors participated in the paper writing and the experimental design. In addition, François Remy and Alfiya Khabibulina worked on the Tatar experiments. Pieter Delobelle worked on the Dutch experiments. Hayastan Avetisyan worked on the Armenian experiments. Finally, Miryam de Lhoneux and Thomas Demeester participated in the ideation process and provided guidance and feedback.

**Acknowledgments**

We thank Matthieu Meeus and Anthony Rathé for kickstarting this line of research with their personal project based on BLOOM and LLama2.

François Remy received the financial support of the Vlaams Agentschap Innoveren & Ondernemen (VLAIO) through its ADAM project. This research also received funding from the Flemish Government under the "Onderzoeksprogramma Artificiële Intelligentie (AI) Vlaanderen" programme, and from the Research Foundation – Flanders (FWO-Vlaanderen) under the project G0C2723N. Pieter Delobelle was also supported by the Research Foundation - Flanders (FWO) under EOS No. 30992574 (VeriLearn) and received a grant from "Interne Fondsen KU Leuven/Internal Funds KU Leuven".

The resources and services used in this work were in part provided by the VSC (Flemish Supercomputer Center), funded by the Research Foundation - Flanders (FWO) and the Flemish Government, and in part by the GPULab, the machine learning infrastructure for AI computing built in collaboration between UGent, UAntwerpen and the imec research and development center.

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

## A    Limitations

Our proposed trans-tokenization strategy, while effective, is not without its limitations.

Firstly, initializing the target language's token embeddings with those from a high-resource language can inadvertently transfer certain cultural and idiomatic patterns from that language to the new one. This may not be desirable, especially when the source and target languages have significant cultural or linguistic differences. However, as the availability of target language training data increases, this issue should tend to diminish.

Secondly, our intra-word many-to-many token mapping approach relies on a left-to-right alignment assumption, which may not always be optimal. In theory, it is possible to extend our method to accommodate different alignment strategies, but this has not been explored in the current study.

Lastly, our fine-tuning process fully utilizes the bottom two and top two layers without employing the Layer-wise Relevance Analysis (LoRA) technique proposed by Hu et al. (2022). This results in a significant VRAM weight for every language supported on a GPU. The use of more efficient adapters, such as LoRA, could have significantly reduced this weight, making our approach more resource-efficient. It might also be possible to train a small projection on top of the existing embedding matrix, to avoid having an entire embedding table per supported language.

## B    Future Work

While our trans-tokenization strategy presents a significant step forward in cross-lingual vocabulary transfer, there are still areas for improvement. We hope that future research will address these limitations, further enhancing the applicability and efficiency of our approach. We also envision a couple of other future works.

Firstly, it would be interesting to investigate the possibility of restoring the mapping after each training epoch during the initial pretraining phase, instead of adding new language at the end like we are proposing in this article. This could potentially enhance the stability and convergence of the training process. Additionally, we aim to explore the integration of the mapping as an ongoing loss during pre-training, which may further improve the quality of the transferred vocabulary (as hinted in Appendix J).

Secondly, we intend to build a Hydra LLM that can support a larger number of languages. This could involve reusing the same fine-tuned layers for all, or families of, languages. By doing so, we aim to increase the resource efficiency of our approach and facilitate the development of inference-friendly LLMs for an even wider range of languages.

Lastly, we hope to see the community develop libraries and infrastructure that enable the efficient use of cross-lingual batch-processing with Hydra LLMs. This would allow for more effective utilization of computational resources, further reducing the data and time requirements for training high-quality models.

In conclusion, our work presents a solid foundation for future research in cross-lingual vocabulary transfer and language adaptation of LLMs. We look forward to the advancements that will be made in this field and the positive impact they will have on the empowerment of languages worldwide.

## C    Release statement

Together with this publication, we release the code and documentation of our trans-tokenizers library, which facilitates the conversion of models from one tokenizer to another. All our trained models will be released on the HuggingFace hub, with the tag "tweety", to enable the community to replicate our work. Finally, we also open source our Tatar summarization dataset on Huggingface.

# D  Relationship with Related Works

In this section, we would like to provide more insights into our methodological choices, in relation to other previously-published approaches.

During the review cycle of this paper, the motivation behind the change of methodology between our earlier BERT-based Tik-to-Tok method (Remy et al., 2023) and this new article has been questioned given the lack of direct comparison between our new method and previous vocabulary transfer methods. We added comparisons with the WECHSEL method (Minixhofer et al., 2022) to address those concerns, but we also wanted to add additional notes gained from previous experiments not part of this article.

While we did not perform apple-to-apple comparisons of Trans-Tokenization with our previous methodology (Tik-to-Tok), we initially applied the latter to the Dutch conversion of LLama2 in collaboration with Matthieu Meeus and Anthony Rathé[4], with significantly worse conversion results than while converting BERT-models with the same technique, which prompted us to develop the Trans-Tokenization method. Since then, we also added several baselines from previous works converting BERT-models, all with unsatisfactory results (see Table 5 and Table 1), and this despite the fact that these methods produce good results for the conversion of BERT-models.

Our current hypothesis is that the key difference between BERT-models and GPT-models lie in the auto-regressive nature of the GPT-models. While BERT-models are frequently used to analyze fully-formed sentences, auto-regressive models quickly suffer from compounding errors, as they require very specific sequences of tokens to be generated in tight succession, at the risk of quickly veering off-domain otherwise due to typos. This means that to be usable, the embeddings of tokens in generative models do not only contain semantic information, but also very specific n-gram modeling information (see Figure 2 below).

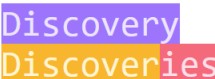 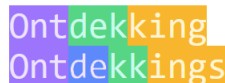

Figure 2: Illustration of the arbitrary nature of token alignment which can be captured by evidence-based SMT mappings (trans-tokenization) but not by character-based mappings.

While the semantic information pertaining to tokens can be recovered well using character-based overlap metrics (because semantically-connected words usually share similar character combinations), the required token-sequence modeling is inherently specific to the exact choices of tokenization in both the source and target languages, and it cannot be recovered without consideration of the precise mappings taking place. This is where our SMT approach shines, because it can align the first token of a word in the source language precisely to the first token of that word translated in the target language, based on token translations. This reduces the number of tokens from which a token is sourced from, and enhances the ability of converted LLMs to generate correct sequences of tokens, even for less frequent combinations.

Many other token mapping techniques, such as FOCUS (Dobler & de Melo, 2023), also make strong assumptions about the accidental or explicit exposure of language models to the target language. For example, FOCUS assumes that mapping a new token to its former constituents will make sense for the language model, but this is not true if the language was never seen at all, or seen rarely. These approaches might however still shine for converting LLMs between high-resources or mid-resources languages. When LLMs are already fluent in a language, adjusting their vocabulary in this way can bring performance gains at inference time, without requiring to also transfer the knowledge from English.

In conclusion, our previous Llama2 experiments and our new baselines show that token mappings based on character-ngrams are not sufficient for GPT-models, where token mappings need to transfer precise information about arbitrary token sequences.

---

[4]Both affiliated with Oqton, a software company accelerating intelligent manufacturing with AI.

# E   Mapping Dutch to English: An example

Trans-tokenizing a model start with a parallel resource between the two languages to map to each other. This resources can be a noisy parallel corpus such as NLLB, or it can be word translation dictionary (although a parallel corpus is preferable).

```
...
I'm only fifteen!              ||| Ik ben pas vijftien!
We saw 15 of them.             ||| Wij zagen er vijftien.
Fifteen maybe?                 ||| Mischien vijftien?
...
```

Listing 1: **Parallel corpus**

This corpus is first tokenized for each language:

```
...
_I ' m _only _fifteen !        ||| _Ik _ben _pas _vijftien !
_We _saw _15 _of _them .       ||| _Wij _zagen _er _vijftien .
_Fif##teen _maybe ?            ||| _Mis##chien _vijftien ?
...
```

Listing 2: **Tokenized corpus**

After using an SMT-based alignment tool, token matching counts are provided:

```
...
13721    _vijftien    _fifteen
12293    _vijftien    _15
544      _vijftien    _Fif##teen
...
```

Listing 3: **FastAlign alignment counts**

After doing the many-to-many token mapping:

```
...
13721    _vijftien    _fifteen
12293    _vijftien    _15
272      _vijftien    _Fif
272      _vijftien    teen
...
```

Listing 4: **Per-token alignment counts**

By normalizing the counts token per token, mapping probabilities can be derived:

```
...
_vijftien := 0.52*_fifteen + 0.46*_15 + 0.01*_Fif + 0.01*teen
...
```

Listing 5: **Final mapping weights**

# F  Details on the SART Word Analogy Task

In this appendix, we detail the SART results per category, and discuss the related findings.

| | SkipG. | FastText | GloVe | Mistral | Mistral+FT | Rand | Base | Tweety | Mistral+GTrans |
|---|---|---|---|---|---|---|---|---|---|
| capital-country | 40.51 | 32.31 | 15.53 | 54.75 | 55.49 | 0.00 | 37.69 | 73.53 | **83.92** |
| country-currency | 4.55 | 5.45 | 5.45 | 59.09 | 47.27 | 0.00 | 30.00 | 62.73 | **90.00** |
| capital-republic-rf | 33.52 | 23.08 | 30.22 | **45.05** | 39.56 | 0.00 | 2.20 | 14.29 | **45.60** |
| man-woman | 40.46 | 38.32 | **41.03** | 1.14 | 12.68 | 0.01 | 6.27 | **41.03** | 26.35 |
| adj-antonym | 8.61 | 7.43 | 6.73 | 0.00 | 3.43 | 0.00 | 13.10 | **49.31** | 22.65 |
| noun-antonym | 8.78 | 7.39 | 7.67 | 0.04 | 2.20 | 0.00 | 7.31 | **53.27** | 28.24 |
| name-occupation | 27.95 | 12.76 | 15.71 | 2.69 | 17.31 | 0.00 | 36.47 | **51.22** | 11.92 |
| **Average:** | 23.48 | 18.11 | 17.48 | 23.25 | 25.42 | 0.00 | 19.00 | **49.34** | 44.1 |

Table 7: Accuracy of models on Tatar the semantic word analogies from the SART dataset, broken down by sub-task. In the main paper, only the average was reported.

We believe that the poor performance of TweetyMistral in the `Capital-of-Russian-Province` to `Russian-Province` test is a good evidence that trans-tokenization is a transfer learning. Our other tests show that Mistral did not master this task in English either.

# G  Details on the Tatar Summarization Task

To evaluate the performance of Tatar models on the text summarization task, we had to generate a suitable dataset (as none existed prior to our work). An important factor for the correct evaluation of summarization is to ensure that the reference summary is not the result of a machine translation process, as this would result in incorrect and simplified language. Therefore, we decided to sample real snippets of text from our training corpus, either one or two sentences long, between 60 and 180 characters, to serve as our summary references.

To generate longer texts based on these seeds, we decide to rely on existing models in English. We therefore translated these snippets into English using Google Translate. Then, we fed those snippets to Mistral Instruct and asked it to generate a longer version of that text. Generations were then evaluated for quality using 3 tests, in order to only include high-quality expansions.

The first test ensured that the expanded text was at least twice as long as the initial text. Shorter expansions were discarded (62% of the generations). The second test ensured that an NLI model could predict with more than 95% certainty that the summary was entailed by the expanded text. Expansions with unclear entailment were discarded (16% of the generations). The third and final test ensured that neither the beginning nor the end of the expanded text were sufficient to entail the seed, meaning that information from the seed was property spread in the entire expanded text. Generations which entailed the see with more than 75% certainty with crops of length smaller than 1.5x the seed were discarded (6% of the generations).

This left around 13% of the generations, or 2179 seed-expansions pairs. The English expansion were then translated back into Tatar using Google Translate. As the expansions do not contribute to the ChrF loss, it is not as important for them to be in native Tatar as it is for the references.

# H  Details on the Tatar Translation Task

For our Short Text and Long Text evaluations, we reused the text summarization dataset we generated previously (see Appendix G). The input provided to the model was the English translation of the seed (or its expansion) produced by Google Translate, and the reference was the seed from which this translation was made. This way, we evaluate the model translations on a real Tatar snipped sampled from the web. As the English inputs do not contribute to the ChrF loss, it is not as important for them to be in native English as it is for the Tatar references.

For our Social Media evaluation, we scraped 125 English snippets from the social network `mastodon.social`, by sampling from the most popular posts from the network on Saturday 2024-03-16. The extracted snippets were manually checked for their ability to be understood in context, the appropriateness of their length, and their exclusive usage of the English language. We also filtered messages pertaining to sensible topics which could cause discomfort to our translator agency (e.g. eroticism, pandemics, armed conflicts).

This resulted in a set of 125 snippets of 60 to 180 characters long. A professional translation agency was then hired to translate these snippets in Tatar, and these translations were used as a reference for the task. We noted, however, similarities between the provided translations and those of Google Translate, which might have been the result of a data contamination (see next appendix).

## I  Analysis of possible Google Translate data contamination

We suspect that the translations provided by the translation agency for the Social Media task were partially contaminated by Google Translate, either directly through inspiration or indirectly through the use of translation memories.

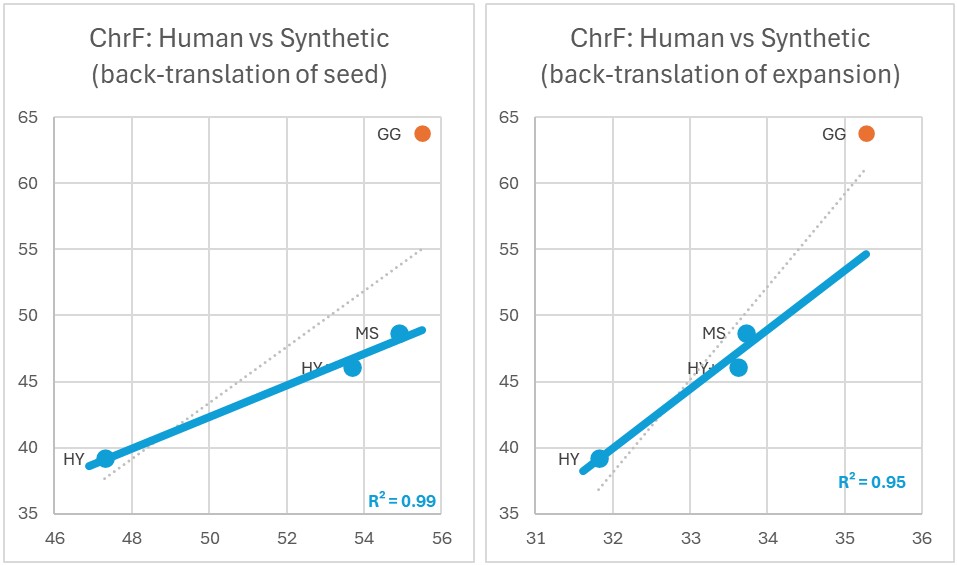

Figure 3: Google Translate results (in orange) are not in line with the otherwise strong cross-task correlations of the other models. We estimate a real score of about 53 instead.

For this reason, we cross the Google Translate result for that experiment, and refrained from providing a "best result" in bold. Based on the correlations found before, we estimate the true score of Google Translate on the Social Media task to be situated between 49 and 55.

## J  Impact of Source Language Choice

While we did not conduct enough experiments to make strong claims about the matter in this paper, we investigated whether the source language from which a mapping was made had a strong influence on the training results. We did this using the TowerInstruct model, which supports English and Russian, two languages for which enough data exists to create high-quality token mappings to Tatar. An interesting aspect of the TowerInstruct model is that each of the 10 languages it supports received the same amount of training data, which should ensure each language is given the same importance by the model.

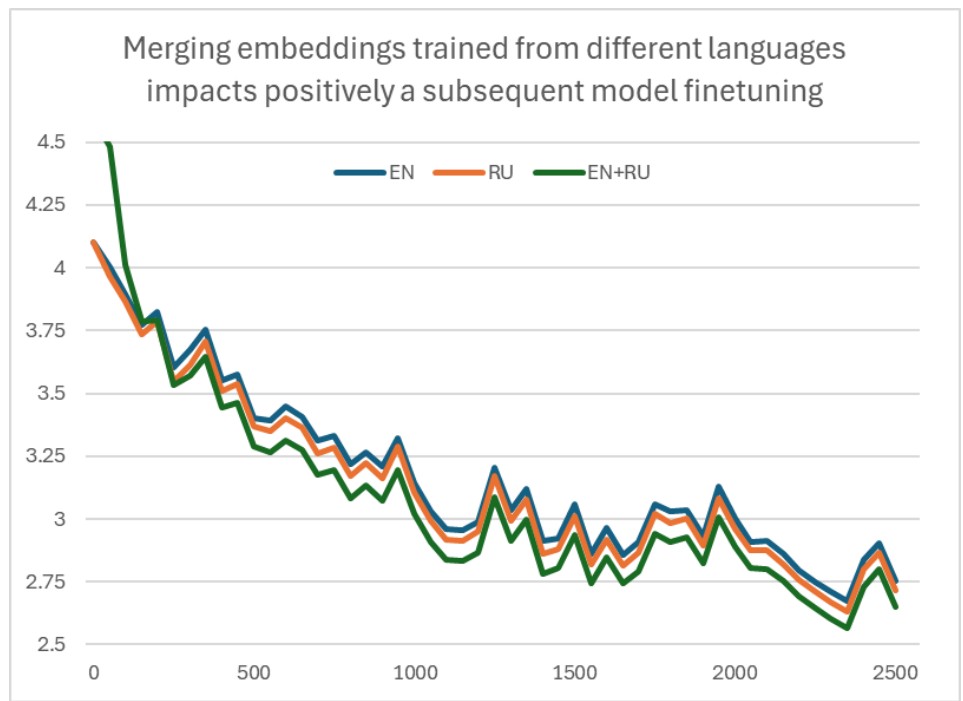

Figure 4: We find that neither the English-to-Tatar nor the Russian-to-Tatar mapping perform better for transfer learning through trans-tokenization. We attribute this to the fact that TowerInstruct being trained with corpus of equal size for English and Russian, and that neither language is particularly close to Tatar. However, combining both initializations provides some benefit.

We also tried merging the models after finetuning the embeddings separately for Russian and English mappings; while this worked, this did not bring additional benefits over merging early, while costing twice the training time.

Intrigued by this finding, we measured the cosine similarity between Russian-initialized and English-initilazed embeddings, and found them to be quite dissimilar (cosine similarity of 0.3). This similairty did not increase meaningfully after finetuning (see Figure 5).

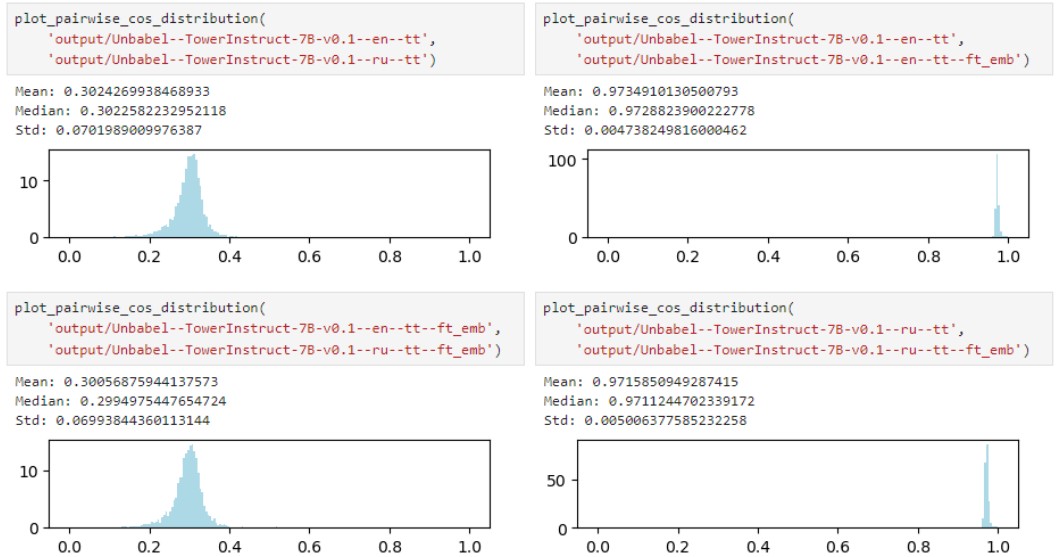

Figure 5: Cosine Similarity Analysis of English-initialized and Russian-initialized embeddings of Tatar tokens, revealing only a very limited degree of similarity.

We hypothesize that the reason for this is that only a small subspace of the TowerInstruct embedding matrix is perceived by the transformer as a result of its projections, and many embeddings would produce the same output through the transformer, while looking substantially different in the full embedding space.

To test this hypothesis, we trained a projection layer using a constrastive strategy, such that the English-initialized and the Russian-initialized embeddings of a token project to the same value, while embeddings of different Tatar tokens remain as different as possible. We were easily able to find such as projection, which potentially confirms our intuition.

Interestingly, this projection can then be applied to the embeddings of tokens from the original vocabulary of TowerInstruct. In our preliminary analysis, the projection appeared effective to bring closer the embeddings of semantically similar tokens across languages (not limited to English and Russian). We however leave the exhaustive analysis of these patterns to a future work, for lack of time and space.

## K    Analysis of the English-Armenian Mapping

The Tweety-Armenian model performed very similarly to the Tatar during and after training, despite the two languages being completely unrelated (see Table below).

| Model | Perplexity | | #Tokens |
|---|---|---|---|
| tweety-7b-armenian-v24a (ours) | | | |
| *(2x2 layers + embed.)* | **7.23** | exp(1.9786) | 123M |
| *(2x2 layers + embed.)* | 8.41 | exp(2.1289) | 107M |
| *(embeddings only)* | 19.55 | exp(2.9732) | 41M |

Table 8: Train perplexity per native token of our TweetyMistral model for Armenian

Due to a lack of readily usable downstream tasks on which to benchmark our Armenian model, a qualitative analysis of the mapping was performed instead, to document the areas that can still be improved.

As expected, the analysis of the English-Armenian word-level mapping revealed substantial differences in the number of unique words between the two languages, with Armenian having more than double the unique words compared to English. This discrepancy is indicative of Armenian's rich morphological structure. Indeed, Armenian is an agglutinative language, meaning that words are often formed by stringing together morphemes to create complex words. This results in a large number of word forms for a single lemma, making it difficult for alignment models to achieve high coverage accurately.

This analysis also revealed the low quality of the parallel corpus used for alignment (in line with the inadequacy of this data for finetuning HydraLLMs highlighted in Table 4, where the parallel data proved detrimental to translation performance, unlike back-translated data). However, most highly-occuring alignments proved semantically correct. Words with the high number of translations (such as "*setting*", "*thread*", "*push*"), and the most translation entropy (such as "*break*", "*up*", and "*pick*") indeed correspond to popular English words exhibiting extreme translation diversity, likely due to their polysemous nature and varied contextual usage. For instance, "*setting*" can translate to multiple Armenian verbs, nouns, and idiomatic expressions, reflecting its contextual flexibility. The token "*break*", for instance, has the highest entropy value (8.3). It has multiple potential translations such as "*kotrel*" (to break), "*yndmijum*" (intermission), and "*cheghkel*" (to crack) [ARM]. These translations correspond to different senses of "*break*", illustrating its semantic range.

Two interesting phenomena also worth studying are the gender-neutral aspect of third-person pronouns in Armenian (where both "*he*" and "*she*" contribute to the token for "*na*" [ARM]) and the polysynthetic nature of Armenian (in particular of negation and internalized subjects, showcased by the Armenian word "*che'in*", a negated form of the verb "*to be*" in the past tense and plural number which encapsulates both the subject ("*they*"), the verb ("*were*"), and the negation ("*not*"), which in English would require multiple words, and whose Armenian embedding is indeed initialized as a mixture of these three components).

One interesting data quality issue that was spotted is that translations in the parallel corpus are not always literal, causing correct sentence alignments to nonetheless contribute to incorrect word alignments. For instance, the translation of "*improve*" to "*barelavum*" is incorrect. "*Barelavum*" is the noun form meaning "*improvement*". It refers to the state or process of improving, not the action itself. The correct translation for the verb "*improve*" should be "*barelavel*", which is the infinitive form of the verb.

Overall, the mapping appeared very usable as a statistical tool, but is not that suitable as a translation dictionary. The mapping is particularly deficient concerning words invovled in idiomatic expressions and phrasal verbs, as the FastAlign model struggles with these due to their contextual dependencies. It sounds likely that better results could be achieved by cleaning the parallel data before computing the word-level mappings and by using a better alignment tool.

# L   Experimental Details

## L.1   Tatar model

We compute the token alignment using the NLLB (NLLB et al., 2022) parallel corpus.

| | |
|---|---|
| **Source model** | `mistralai/Mistral-7B-Instruct-v0.2` |
| **Source language** | `en` |
| **Target language** | `tt` |
| **Target tokenizer** | `new(BPE, 32k, oscar-corpus/OSCAR-2301[tt])` |
| **Parallel data** | `NLLB[en-tt]` |
| **Alignment unit** | `PREFER-WORDS` |
| **Alignment min count** | `10` |

Table 9: Hyperparameters of the token mapping of our Tatar model.

We finetune the embeddings on the first 41M tokens of OSCAR (Ortiz Suárez et al., 2019).

| | |
|---|---|
| **Init model** | `mistralai--Mistral-7B-Instruct-v0.2--tt` |
| **Train data** | `oscar-corpus/OSCAR-2301[tt]` |
| **Trained layers** | `embeds, lm_head` |
| **GPU** | `1 x NVIDIA A100 80Gb` |
| **GPU Time** | `4 GPU hours` |
| **Seq size** | `512 tokens` |
| **Batch size** | `32 (8x4)` |
| **Max Steps** | `2500` |
| **LR Schedule** | `constant_with_warmup` |
| **LR Peak** | `2e-5` |
| **Warmup** | `75 steps` |

Table 10: Hyperparameters of the embedding finetuning of our Tatar model.

We then unfreeze 2x2 layers on the next 66M tokens of OSCAR (Ortiz Suárez et al., 2019).

| | |
|---|---|
| **Init model** | `mistralai--Mistral-7B-Instruct-v0.2--tt--ft_emb` |
| **Train data** | `oscar-corpus/OSCAR-2301[tt]` |
| **Trained layers** | `embeds, layers[0,1,30,31], lm_head` |
| **GPU** | `1 x NVIDIA A100 80Gb` |
| **GPU Time** | `7 GPU hours` |
| **Seq size** | `512 tokens` |
| **Batch size** | `32 (8x4)` |
| **Max Steps** | `4000` |
| **LR Schedule** | `linear_with_warmup (assuming max_steps=7500)` |
| **LR Peak** | `2e-5` |
| **Warmup** | `75 steps` |

Table 11: Hyperparameters of the 2x2+E finetuning of our Tatar model.

### L.2 Armenian model

We compute the token alignment using the NLLB (NLLB et al., 2022) parallel corpus.

| | |
|---|---|
| **Source model** | `mistralai/Mistral-7B-v0.1` |
| **Source language** | `en` |
| **Target language** | `hy` |
| **Target tokenizer** | `new(BPE, 32k, oscar-corpus/OSCAR-2301[hy])` |
| **Parallel data** | `NLLB[en-hy]` |
| **Alignment unit** | `PREFER-WORDS` |
| **Alignment min count** | `10` |

Table 12: Hyperparameters of the token mapping of our Tatar model.

We finetune the embeddings on the first 41M tokens of OSCAR (Ortiz Suárez et al., 2019).

| | |
|---|---|
| **Init model** | `mistralai--Mistral-7B-Instruct-v0.2--hy` |
| **Train data** | `oscar-corpus/OSCAR-2301[hy]` |
| **Trained layers** | `embeds, lm_head` |
| **GPU** | `1 x NVIDIA A100 80Gb` |
| **GPU Time** | `4 GPU hours` |
| **Seq size** | `512 tokens` |
| **Batch size** | `32 (8x4)` |
| **Max Steps** | `2500` |
| **LR Schedule** | `constant_with_warmup` |
| **LR Peak** | `2e-5` |
| **Warmup** | `75 steps` |

Table 13: Hyperparameters of the embedding finetuning of our Armenian model.

We then unfreeze 2x2 layers on the next 82M tokens of OSCAR (Ortiz Suárez et al., 2019).

| | |
|---|---|
| **Init model** | `mistralai--Mistral-7B-Instruct-v0.2--hy--ft_emb` |
| **Train data** | `oscar-corpus/OSCAR-2301[hy]` |
| **Trained layers** | `embeds, layers[0,1,30,31], lm_head` |
| **GPU** | `1 x NVIDIA A100 80Gb` |
| **GPU Time** | `13 GPU hours` |
| **Seq size** | `512 tokens` |
| **Batch size** | `32 (8x4)` |
| **Max Steps** | `7500` |
| **LR Schedule** | `linear_with_warmup (assuming max_steps=7500)` |
| **LR Peak** | `2e-5` |
| **Warmup** | `75 steps` |

Table 14: Hyperparameters of the 2x2+E finetuning of our Armenian model.

### L.3 Dutch model

We compute the token alignment using the concatenation of two parallel corpora: Open Subtitles (Lison et al., 2018) and NLLB (NLLB et al., 2022).

| | |
|---|---|
| **Source model** | `mistralai/Mistral-7B-v0.1` |
| **Source language** | `en` |
| **Target language** | `nl` |
| **Target tokenizer** | `yhavinga/gpt-neo-1.3B-dutch` |
| **Parallel data** | `OpenSubtitles2018[en-nl] + NLLB[en-nl]` |
| **Alignment unit** | `PREFER-WORDS` |
| **Alignment min count** | `20` |

Table 15: Hyperparameters of the token mapping of our Tatar model.

We train and evaluate our model on a cleaned version of the Dutch fraction of C4[5]. For the finetuning, we use 2 A100 GPUs for a total of 40 GPU-hours, with an effective batch size of 256 and a maximal context length of 8,192.

| | |
|---|---|
| **Init model** | `mistralai--Mistral-7B-v0.1--nl` |
| **Train data** | `yhavinga/mc4_nl_cleaned` |
| **Trained layers** | `all` |
| **GPU** | `2 x NVIDIA A100 80Gb` |
| **GPU Time** | `400M tokens: 40 GPU hours (2x20)` |
| | `8.5B tokens: 1k GPU hours (4x250)` |
| **Seq size** | `8192 tokens` |
| **Batch size** | `256 (8x16x2)` |
| **Epochs** | `1, but stopped early` |
| **LR Schedule** | `linear_with_warmup` |
| **LR Peak** | `1e-4` |
| **Warmup** | `300 steps` |

Table 16: Hyperparameters of the full finetuning of our Dutch model.

---

[5]https://huggingface.co/datasets/yhavinga/mc4_nl_cleaned

## M Origins of the Tweeties

The name Tweety comes from the abbreviation of our proposed method, trans-tokenization (TT for short). The name sounds pleasing to hear, and is semantically associated with a bird. This association made the choice of a mascot easy.

To enable each model to have its own personality and brand, we developed a template providing space for a flag and a background photograph, which helps locate the language and the region of the world it is usually spoken in.

This strong association between a language, a country, and a location is of course very incomplete, as many languages are spoken in several regions worldwide. However, we envision that the low computation cost of training new trans-tokenized models would enable dialect-specific LLMs in the future, helping solve issues in cases where a language is spoken in more than one country or region.

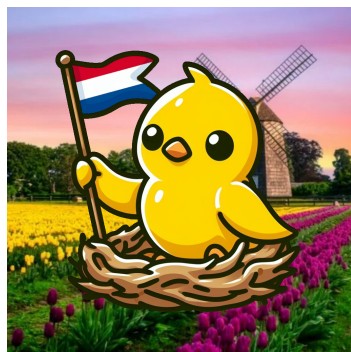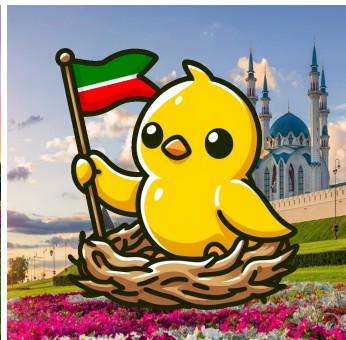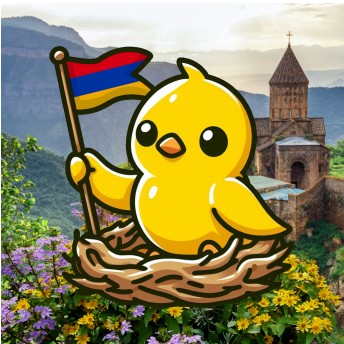

**Tweety Dutch, Tweety Tatar, and Tweety Armenian.**

*Our first three Tweeties.*

