# OpenReview forum: "Trans-Tokenization and Cross-lingual Vocabulary Transfers: Language Adaptation of LLMs for Low-Resource NLP"
_colmweb.org/COLM/2024/Conference — COLM_

### Official Review · Reviewer_opdP · 2024-05-09

**Rating:** 6
**Confidence:** 4
**Ethics Flag:** 1

**Summary:**

This paper presents a new method for adapting a pretrained langauge model to new languages by using "translation tokenization": an SMT system learns an alignment between two languages (one old and one "new"), and this alignment is used to initialize embeddings for tokens of the new language based on an LMs embeddings for the old language.  They also use "Hydra" LMs, which have multiple embedding tables and LM output heads as a way of doing multilingual understanding and translation.  Many results are reported (language modeling for various languages of different resource-ness, translation, etc), but are somewhat under-discussed, which makes it hard to tell exactly what has been learned from this paper.

**Questions To Authors:**

## General / expository questions

* The introduction mentions zero-shot translation for the Tatar language.  But the method for tokenization requires prallel data to generate a token alignment.  It seems misleading to call it truly zero-shot given the use of parallel data in the learning of new embeddings.  To put it differently: trans-tokenization relies on translation, so a system which uses it doesn't seem to be a truly zero-shot translation system.  It does not "completely bypassing the need for high-quality parallel data" (abstract), since that's needed for the tokenization.

* Since you are using a parallel corpus only for token alignment, have you compared to having a bilingual dictionary?

* In the other direction: you compare your Hydra models to ones also using BackTranslation; why not also compare to fine-tuning on translation data, which you do have even for Tatar?  This would be a fairer comparison than MS and Google Translate.

* "We hypothesize (but did not verify) that the two bottom layers of the source model should probably be used to encode tokens from the original vocabulary instead of the layers finetuned for the target language. However, in our experiments, we always used the finetuned layers from the trans-tokenized model, as this did not seem to cause any issue."  This felt odd and out of place, and was hard to decipher at the point it occurred in the text.  I would either expand with more details or remove it entirely.

* I would have appreciated more detail and discussion in the results sections.  For instance, we aren't told what to learn from Tables 1 and 2 at all (and Table 2 is never even referenced).  It seems interesting and informative in Table 1 that MistralFT is better than TweetyMistral when _only_ the embeddings are trained (and not the 2x2 layers), but this isn't mentioned or discussed at all.

## Related Work

* On adapting embeddings for new vocab: https://aclanthology.org/2023.emnlp-main.829 , https://aclanthology.org/2023.mrl-1.20

**Reasons To Accept:**

* Adapting language models to low-resource languages with new tokenization is a very important problem and an interesting method is proposed here (which does, however, require parallel data).
* "Hydra" language models are a potentially useful way of adapting to new languages.
* A large set of experiments.

**Reasons To Reject:**

* Extremely under-reported results.  There's a plethora of results tables, but no summaries of what they show us and what we learn from them, and no real discussion section either.  I think there's a lot of great work in this paper, but the authors should spend more time helping the reader extract insights from all of their experiments.
* Missing comparisons and baselines: for Tatar translation, for instance, they compare a "zero-shot" (more on this term in the questions below) system, to one with backtranslation, to two commercial systems.  There should also be a comparison to a system that uses their parallel data for fine-tuning instead of just back-translation.
* It would also be valuable to compare the trans-tokenization method to several other token re-initialization schemes from the literature (e.g. see "Related Work" below), especially since those don't require parallel data.

---

> ### Author Rebuttal · Authors · 2024-05-29
>
> **I think there's a lot of great work in this paper, but the authors should spend more time helping the reader extract insights from all of their experiments.**
>
> Thank you for the compliment. In retrospect, we agree that a deeper discussion of the results would have helped our paper greatly. We will leverage our extra page to provide an insightful discussion section that is well-informed thanks to our discussions.
>
> **There should also be a comparison to a system that uses their parallel data for fine-tuning instead of just back-translation**
>
> Based on your feedback, we trained several new baselines, and were able to show that our method outperforms finetuning TowerInstruct on the parallel data, and even that HydraTower degrades when finetuned on the parallel data. In line with the PCs clarifications regarding rebuttals, we link to limited updated content in Table 5:
>
> https://imgur.com/BV0Mj9T
>
> This is not surprising for low-resource languages, where the data is often scraped and of poor quality. For instance, the Tatar NLLB corpus contains many mismatched entries, and almost ~60% of religious texts. This is a key advantage of our method: while our initialization does require some form of parallel data, it does not necessitate high-quality parallel data. Even a word translation dictionary suffices. Our approach uses parallel data as a weak initialization and learns the target language from monolingual data alone, which is usually of better quality.
>
> **Comparison to Bilingual Dictionary:**
>
> During the development, we have applied our conversion strategy to produce a state-of-the-art Frisian RoBERTa model based on a Dutch RoBERTa model using only a translation dictionary as parallel data. You can find more about this in the supplementary materials, especially the Dutch-to-Frisan experiments. However, this performs slightly worse than using SMT-based alignment. Some words are polysemic and have multiple translations, and a dictionary does not provide an accurate estimate of how often each sense occurs in real texts. Based on the experience acquired from that experiment, we decided not to pursue dictionary-only mapping in this paper.
>
> **Comparison to Other Token Re-Initialization Schemes:**
>
> Based on the feedback from another reviewer, we also evaluated MaLA and WECHSEL (a dictionary-based method) and found that our initialization outperformed theirs by a very significant margin (see rebuttal of pLrB). We will include a discussion of these points.

---

> > ### Comment · Reviewer_opdP · 2024-06-05
> > **Thanks; promising revisions**
> >
> > I thank the authors for their detailed and serious engagement with my review.
> >
> > Other re-initialization scheme: in response to pLrB, they mention taht FOCUS only applies to MLMs (multilingual models).  While that was the focus of that paper, I do believe the method would work just the same in the current paper's setting and so could be another good comparison.
> >
> > A lot of my comments and their replies require a non-trivial amount of writing, e.g. improved results discussion and the inclusion of a new results table.  I do think these will benefit the paper a lot and probably can be done with one extra page, but I am hesitant to increase my score at the moment since they are not yet done and I already put it above acceptance threshold.

---

> > > ### Author Response · Authors · 2024-06-06
> > >
> > > Thank you for your thoughtful consideration of our work. We really appreciate your feedback.
> > >
> > > Regarding the comparison to FOCUS, it is essential to note that FOCUS is specifically designed for multilingual Masked Language Models (MLMs) and would require significant adaptation to be applicable to LLMs like ours. We understand your curiosity on this, but our past experiments have shown that methods similar to FOCUS usually perform significantly worse in this context.
> > >
> > > Moreover, FOCUS is not applicable to the Tatar language due to its dependence on XLM-R, which does not support Tatar at all. FOCUS initializes embeddings based on tokenizer overlap, as illustrated in Fig. 1 of their paper, which does not work for unsupported languages. This highlights a significant advantage of our method, which requires no prior model supporting the target language of the conversion.
> > >
> > > Additionally, we have already provided an updated Table 5 (https://imgur.com/BV0Mj9T) in our previous submission. The updated PC's guidelines however prevent us from including long pieces of text. Nonetheless, we are confident that the additional space we have will be utilized effectively for further discussion and elaboration on our results, and will contribute to an interesting presentation at COLM.
> > >
> > > We appreciate your understanding and hope these clarifications will address your concerns.

---

### Official Review · Reviewer_hY6q · 2024-05-10

**Rating:** 6
**Confidence:** 4
**Ethics Flag:** 1

**Summary:**

This paper is about using a separate tokenizer per language under the multilingual scenario.
Since the tokenizer most current LLMs are depending on, i.e. splitting a word into subwords based on their frequencies, is highly biased toward frequent languages, it may harm the performance for low-resource languages.
This work suggests training a tokenizer for each language; which the authors coin "trans-tokenization."
One challenge that the trans-tokenization approach should overcome is adapting the token embeddings pre-trained from specific (usually frequent) languages into low-resource one, since for low-resource languages there hardly exist enough data to obtain high-quality token embeddings.

The authors try to address the challenge through the use of parallel corpora.
Using off-the-shelf statistical machine translation (SMT) tool, it finds the mappings between word (or token) in one language and another language.
Based on the frequencies of mappings for a specific word or token, a token embedding for a low-resource language is obtained via weighted-summing the source token embeddings.

For the evaluation, Tatar language is mainly used as a low-resource language while adopting Mistral as a base LLM.
The evaluation is done on language modeling, language understanding, text summarization, and machine translation.
For baselines, a model without tuning (w/ original tokenizer; Mistral) and with fine-tuning (w/ original tokenizer; MistralFT, w/ language-specific tokenizer + random initialization; MistralRAND, and w/ language-specific tokenizer + initializing embeddings for tokens in both languages with the original ones; MistralBASE) are used.
The proposed model, TweetyMistral, is a model with fine-tuning, where embeddings are initialized based on trans-tokenization.

On low-resource evaluations, the TweetyMistral models generally brought performance gain.
From language modeling experiments, the TweetMistral models whose embeddings and the first and last two layers are fine-tuned performed the best, while MistralFT slightly underperformed TweetMistral ones.
In language understanding, text summarization, and machine translation experiments, the authors tested an interesting baseline: using off-the-shelf translation applications like Google Translate and Microsoft Translator.
Though in text summarization and machine translation the performance of using translation applications performed the best.

The authors also evaluated whether the trans-tokenization approach can also improve mid-resource languages.
In language modeling experiments with Dutch language, the proposed algorithm performed the best, even with much fewer training tokens and even compared to GPT-Neo model trained on Dutch with the same tokenizer.
In language understanding experiments with Dutch, the trans-tokenization approach outperformed other models on 1-shot and 2-shot setups while on 0-shot setup the original Mistral performed the best.

Overall, the paper is well-organized and accompanies a wide coverage of experiments.
Configurations of each experiment are presented in detail in appendices, which will help replicating the work.
Though some questions about the improvements remain, I believe this work will give insights to those who want to transfer a model trained on frequent languages into other languages.

**Questions To Authors:**

1. How is the performance of mid-resource language understanding, summarization, or translation, especially when compared to Mistral+GTrans? In my opinion part of the reason that TweetyMistral underperforms low-resource language understanding is due to its extremely low similarity to English, which may lead to incorrect mappings via SMT.
2. In summarization evaluation, Mistral Instruct is used in making texts longer; did it add any important context about summarization, by using what it already knows, which may change the true answer for the summarization task?
3. Google Translate is already used in generating short summarization data. Could it affect the performance of short text translation evaluation scores, like the data contamination appeared in social media section?

Typo: SQAD → SQuAD

**Reasons To Accept:**

- The paper is organized well and easy to read
- Extensive experiments, with detailed presentation about configurations

**Reasons To Reject:**

- The advantage over the original LLM without language-specific tokenizer is unclear. In most experiments MistralFT or the original Mistral model
- It still requires parallel corpora.

---

> ### Author Rebuttal · Authors · 2024-05-29
>
> **Advantage over the original LLM without language-specific tokenizer:**
>
> For mid-resource languages like Dutch, our method offers a cost-effective alternative to full pre-training, providing the benefits of a language-specific tokenizer. This results in fewer tokens per sequence, more meaningful embeddings, and better assurance that the generations are in proper Dutch.
>
> For low-resource languages like Tatar, Mistral does not work at all. Our method enabled the grounded generation of Tatar text with a very limited training set, a significant step forward in the field.
>
> **It still requires parallel corpora:**
>
> While our initialization indeed requires some form of parallel data, our abstract mentions that our approach does not need **high-quality** parallel data. Even a word translation dictionary suffices for a weak initialization and we learn the target language from monolingual data alone. You can find more about this in the supplementary materials, especially the Dutch-to-Frisan experiments.
> We also ran additional experiments to showcase this, by finetuning our Hydra model on the parallel data, which resulted in a degraded performance (!). You can view this baseline and a few others in our updated Table 5:
>
> https://imgur.com/BV0Mj9T
>
> **Did Mistral add any important context about summarization?**
>
> The summarization task was evaluated on a generated test set where we indeed expand the paragraphs with Mistral (in English). However, to ensure the expansion quality and to prevent as much as possible any undesired expansions, we tested the entailment of the expanded text and their seed during the dataset creation. We believe that this resulted in a solid evaluation, considering the lack of benchmarks in Tatar.
>
> **TweetyMistral underperforms low-resource language understanding is due to its extremely low similarity to English, which may lead to incorrect mappings via SMT**
>
> On the NLU task, our method outperforms all others (inc. Mistral+GTrans). Perhaps the reviewer meant the summarization task, where our method performs similarly* to Mistral+GTrans, but outperforms Mistral and other baselines. This result might indeed be affected by challenges in morphology and grammar. Nevertheless, our model performs as well in Tatar as an English model summarizing (translated) English paragraphs, which we consider a very positive result for a transfer learning method.
>
> [*] The difference between both models on the summarization task is not significant (\sigma=1.5).

---

### Official Review · Reviewer_pLrB · 2024-05-14

**Rating:** 6
**Confidence:** 4
**Ethics Flag:** 1

**Summary:**

On a high level the aim of this work is to offer technology that allows for adapting pretrained LLMs to low-resource languages. One part of the adaptation stems from the adaptation of the tokenizers (the so-called trans-tokenization technique which is the first contribution of the work), and the other part comes through the methodological contribution: the authors introduce the so-called HydraLLMs with multiple swappable language modeling heads and embedding tables. The paper experiments on several languages that come with different levels of resources (Tatar being a case study of the lowest-resource language), and demonstrate that trans-tokenization + HydraLLMs can offer good performance even for languages for which high-quality parallel data cannot be guaranteed.

**Questions To Authors:**

Some clarification on how zero-shot usage of HydraLLM is enabled for Tatar is needed. How is this zero-shot and without parallel data if on Page 6 it is said that "(...) initializing Tatar tokens by averaging mappings from English-Tatar and Russian-Tatar parallel corpora"?

- Is parallel data for trans-tokenisation required or could the embeddings also be initialised via auxiliary cross-lingual word embedding spaces such as done e.g., by the WECHSEL model of Minixhofer et al. (NAACL 2022)?

- Low-resource NLU experiment is based only on Word Analogies - I am not sure that Word Analogies qualify as an NLU task at all. Ideally, another task should be added here.

- It would be interesting (and useful to claim that the Hydra idea is pretty general) to build HydraLLM also atop of some other LLM beyond the TowerInstruct model. Some justification on why exactly TowerInstruct was selected as the base model would also be useful to better depict the experimental setup.

**Reasons To Accept:**

- The work is timely and aligns well with several important LLM-related topics such as: low-data adaptation of LLMs to different (low-resource) languages, (re)tokenization without extensive continued pretraining, massively multilingual LLMs

- The chosen experimental tasks and languages are (for most part) nicely selected and well justified (although some additional experiments with languages with unseen scripts would be required), and...

- The results indicate usefulness of the proposed techniques; some ideas from this work might inspire future research on low-resource adaptation of LLMs.

**Reasons To Reject:**

- My main concern with this work is that ignores a body of very relevant work on the topic of (re)tokenisation and trans-tokenisation which also conducted adjustment of tokenisers to very similar heuristics. That work is not discussed at all nor compared against, which severely hinders my ability to isolate the core contribution of this work in comparison to what was done in prior work (which also had multilingual applications in focus). For instance, all of these papers used some sort of heuristics for smart embedding initialisation in the target language:
*https://arxiv.org/abs/2311.08849
*https://arxiv.org/abs/2305.14481
*https://aclanthology.org/2022.naacl-main.293/
*https://aclanthology.org/2021.emnlp-main.800/
Put simply, the coverage of related work (and therefore relevant and strong baselines) is inadequate, and requires much more attention in the paper.

- Along the same line, there have been other recent attempts towards adaptation of LLMs for low-resource languages, which also have not been cited nor discussed, such as:
*Again the OFA work (https://arxiv.org/abs/2311.08849)
*MaLa-500 (https://arxiv.org/abs/2401.13303)
*https://arxiv.org/abs/2112.10668
This paper again doesn't provide an adequate overview of this line of work.

- Some clarifications on experimental details are needed (see Questions to Authors)

---

> ### Author Rebuttal · Authors · 2024-05-29
>
> **Related Work**
>
> We acknowledge the importance of situating our work within the broader research context. But please note that our focus is on creating high-quality **monolingual** LLMs.
>
> For instance, MaLA-500 is a **multilingual** method and that has a trade-off between supporting more languages and providing better performance for each. We evaluated MaLA-500 on Dutch and found its perplexity a lot higher (18.9) compared to ours (11.1) while using 3B more parameters.
>
> This is also the case UNKs Everywhere, using mBERT. Additionally, OFA and FOCUS also focus on MLM adaptation. This does not apply directly as it would require additional work for autoregressive LLMs (tied weights, etc...).
>
> WECHSEL is indeed very related. We ran evaluations to provide a comprehensive comparison for Dutch using the WECHSEL codebase, and pretrained a mapped Mistral model for ~100M tokens (see https://imgur.com/a/olsIpXL with WECHSEL=highest curve). WECHSEL’s perplexity and loss are magnitudes higher, likely due to lack of mapping for the output weights and the use of a small word-based parallel corpus. We will compare this in-depth in the final manuscript.
>
> Also note that many of these references are recent publications (e.g. NAACL2024) or ArXiv preprints, showing the timely nature of our work.
>
> **Parallel Data**
>
> Parallel texts are indeed not required for trans-tokenization. We tested a translation dictionary for initializing a Frisian RoBERTa based on Dutch RoBERTa. However, our SMT-based alignment performed better thanks to a more accurate initialization (e.g. in case of polysemy). We will include this in the paper, as requested by HhZh.
>
> **NLU**
>
> We agree that additional NLU tasks would strengthen our evaluation. However, the scarcity of benchmarks for truly low-resource languages like Tatar limits our options. We have made significant efforts to develop high-quality Tatar benchmarks, and we hope these contributions demonstrate our commitment to thorough evaluation.
>
> **Hydra models**
>
> While not mentioned in the paper, we have also built a Hydra model on top of Mistral, which can provide answers in fluent Tatar to questions written in English. Due to the lack of quantitative benchmarks for this task, we focused on translation. TowerInstruct is already fine-tuned for translation.
>
> It makes sense to clarify this in the paper. We will report our results on the Mistral Hydra model in an appendix and release that model if that can help boost confidence in our work.

---

> > ### Comment · Reviewer_pLrB · 2024-06-04
> > **Thank you for the response!**
> >
> > The response with the new results clarified some of my main concerns and questions (e.g., I'm glad that a preliminary comparison to WECHSEL has been done, and I hope to see a more detailed comparison in the paper). Other additional experiments (e.g., Hydra atop Mistral) will also strengthen the contribution of the work.
> >
> > One major thing that's still currently lacking is a wider experimental exploration, with more tasks and more languages, and I would like to see performance on the model also on some standard NLU benchmarks with a selection of languages from those benchmarks taken as additional case study.
> >
> > It also remains unclear why parallel data might decrease performance (as written in the response to another reviewer), and if parallel data is still needed for the adaptation of distant target languages (where Dutch-to-Frisian might not be the prime example).
> >
> > I would suggest to add the discussion on massively multilingual models versus monolingual adaptation also to the main paper.
> >
> > All in all, in light of the new information provided in the response, I am happy to adjust my score.

---

### Official Review · Reviewer_HhZh · 2024-05-15

**Rating:** 6
**Confidence:** 3
**Ethics Flag:** 1

**Summary:**

This paper proposes a cross-lingual vocabulary transfer strategy, called trans-tokenization, in order to adapt monolingual large language models for low and mid-resource languages. They make token alignment using a SMT-based alignment tool, to establish a probabilistic token mapping. Then, they perform embedding mapping. Their proposed models experimentally show the competitive performance on various downstream tasks. Moreover, they design a Hydra LLMs base on the multilingual model of TowerInstruct, and demonstrate the state-of-the-art translation model for Tartar language.

**Questions To Authors:**

- Have you ever tried out this approach against non-Latin script or languages linguistically different from English to assess the robustness  of your proposed approach? How much impact does this alignment tool make?

- By introducing the proposed trans-tokenization, how much does tokenized length get changed?

- Since you use the parallel data for mapping, another baseline should be Mistral model finetuned on the parallel data so both model see the same data during training.

- Did you conduct any ablation study to figure out the key factor in your approach?

typos
- "we proposed in a preprint (2023, anonymized) a novel strategy" Are you citing this paper or another paper?
- n-gram -> $n$-gram

**Reasons To Accept:**

- This paper tackles a tokenization adaptation problem. The proposed approach could be applicable to any types of large language model, and it would enhance the multilingual capability.

- Their experimental results on Tartar, the low-resource language, look promising.

**Reasons To Reject:**

- The current manuscript is redundant. The abstract and some sections could be reorganized in order to make some space.

- The results are yet to be technically sound. This paper tackles a problem of cross-lingual vocabulary transfer for low and mid-resource languages, although the experimental results are carried out a few of language sets. This paper could be improved with additional results on more diverse low/mid-resource languages. I am curious if any trends can be observed between English and non-latin scripts. Deep analyses would be also useful.

---

> ### Author Rebuttal · Authors · 2024-05-29
>
> **Have you ever tried out this approach against non-Latin script or languages linguistically different from English to assess the robustness of your proposed approach?**
>
> Two of our experiments use languages written in non-Latin scripts. Modern Tatar is written using a variant of the Cyrilic script including a few letters unique to Tatar, while Ancient Tatar is written in an Arabic script. Additionally, Armenian is written using a script totally unique to the language. Both languages are extremely challenging for LLMs: not even GPT-4o speaks Tatar fluently, and neither language had any form of support in GPT 3.5. We'll make sure to make this information more prominent in the paper by providing examples of the training material.
>
> **How much does tokenized length get changed?**
>
> Our trans-tokenization method significantly reduces the tokenized length. For example, TweetyTatar uses almost 3x fewer tokens than Mistral, as shown in Table 2. This translates into 3x cheaper and 3x faster text generation, and a 3x larger effective context window. Similarly, TweetyDutch uses 33% fewer tokens than Mistral. We will emphasize these benefits more clearly in the final manuscript to highlight the efficiency gains of our approach.
>
> **Since you use the parallel data for mapping, another baseline should be Mistral model finetuned on the parallel data so both models see the same data during training.**
>
> We appreciate your suggestion to include a baseline where the base model is fine-tuned on the parallel data. We conducted this experiment and found that fine-tuning HydraTower on this data degraded its performance (from 47.3 chrF down to 39.6 chrF on the Tatar short text experiment). You can view this baseline and a few others in our updated Table 5:
>
> https://imgur.com/BV0Mj9T
>
> **Ablation study**
>
> We conducted several ablation studies during the development of our method before turning to SMT-based alignment. Initially, we experimented with a different token alignment approach leveraging a multilingual embedding space. However, this introduced many arbitrary hyperparameters and our current method yielded better performance. You can find more about this in the supplementary materials, especially the Dutch-to-Frisan experiments. We will clarify this in the final manuscript to highlight the improvements and the novelty of our current approach.
>
> We hope this rebuttal has adequately addressed your concerns, and we look forward to continuing this discussion.

---

> > ### Comment · Reviewer_HhZh · 2024-06-05
> > **Thank you for the response.**
> >
> > The response clarified my questions so I increased the rating to "6: Marginally above acceptance threshold".

---

> > > ### Author Response · Authors · 2024-06-06
> > >
> > > Thank you for increasing the score to "marginally above acceptance threshold". Given that we provided an additional experiment on parallel data and included 2 non-Latin scripts, are there any other aspects you would like us to address to increase our score to 7 (accept)?

---

### Author Response · Authors · 2024-06-03
**Dear Reviewers, please consider these rebuttals**

Dear Reviewers,

Our team took a significant amount of time to write personalized rebuttals, often bringing up new experimental results to address your questions.

We would greatly appreciate your feedback on our rebuttals.

Thank you for your continued involvment,
The Authors

---

### Decision · Program_Chairs · 2024-07-10

**Decision:**

Accept

**Comment:**

The paper proposes an embedding initialization method for adapting monolingual LLMs to low-resource languages. A token mapping is constructed from FastAlign word alignments over a parallel corpus; this is used to map both input and language modelling head embeddings. The approach is evaluated on transfer to Tartar as a low-resource language and to Dutch as a higher-resource language. One limitation is the lack of comparison to other recently proposed cross-lingual embedding initialization methods, even though some of those were proposed for multilingual adaptation vs monolingual adaptation. Some reviewers also requested more extensive evaluations and additional analysis of the results (of which the authors provided some during the discussion period). However despite these limitations this is a timely paper, so while the authors are advised to include the additional results and analysis in the final version of the paper, the contributions are sufficient to recommend acceptance.